# MEGA: Memory-Efficient 4D Gaussian Splatting for Dynamic Scenes

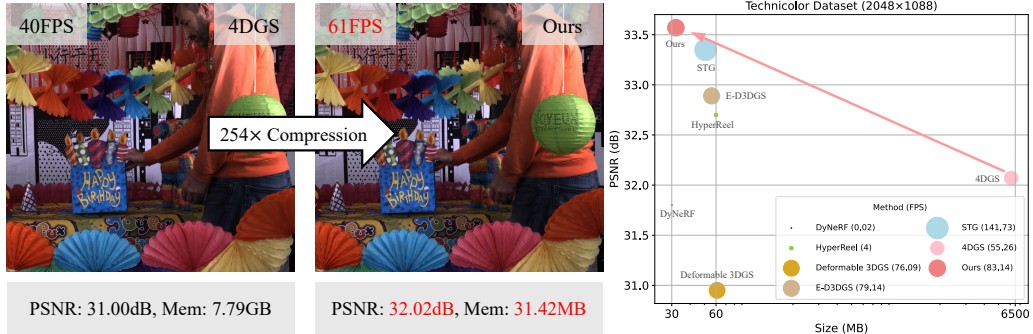

(a) High performance at the *Birthday* scene.    (b) Comparison on quality, size, and speed.

Figure 1: Our approach significantly reduces storage requirements while maintaining comparable photorealistic quality and real-time rendering speed with 4D Gaussian Splatting (4DGS) (Yang et al., 2024a). The core idea is to develop a memory-efficient 4D Gaussian representation and use as few Gaussians as possible to fit dynamic scenes well. (a) 4DGS requires up to 13 million Gaussians to render the *Birthday* scene, whereas our method only needs 0.91 million Gaussians. (b) Quantitative comparisons of rendering quality, storage size, and speed against various competitive baselines on the Technicolor dataset.

## ABSTRACT

4D Gaussian Splatting (4DGS) has recently emerged as a promising technique for capturing complex dynamic 3D scenes with high fidelity. It utilizes a 4D Gaussian representation and a GPU-friendly rasterizer, enabling rapid rendering speeds. Despite its advantages, 4DGS faces significant challenges, notably the requirement of millions of 4D Gaussians, each with extensive associated attributes, leading to substantial memory and storage cost. This paper introduces a memory-efficient framework for 4DGS. We streamline the color attribute by decomposing it into a per-Gaussian direct color component with only 3 parameters and a shared lightweight alternating current color predictor. This approach eliminates the need for spherical harmonics coefficients, which typically involve up to 144 parameters in classic 4DGS, thereby creating a memory-efficient 4D Gaussian representation. Furthermore, we introduce an entropy-constrained Gaussian deformation technique that uses a deformation field to expand the action range of each Gaussian and integrates an opacity-based entropy loss to limit the number of Gaussians, thus forcing our model to use as few Gaussians as possible to fit a dynamic scene well. With simple half-precision storage and zip compression, our framework achieves a storage reduction by approximately 190× and 125× on the Technicolor and Neural 3D Video datasets, respectively, compared to the original 4DGS. Meanwhile, it maintains comparable rendering speeds and scene representation quality, setting a new standard in the field.

## 1 INTRODUCTION

Dynamic scene reconstruction from multi-view videos is gaining widespread interest in computer vision and graphics due to its broad applications in virtual reality (VR), augmented reality (AR), and

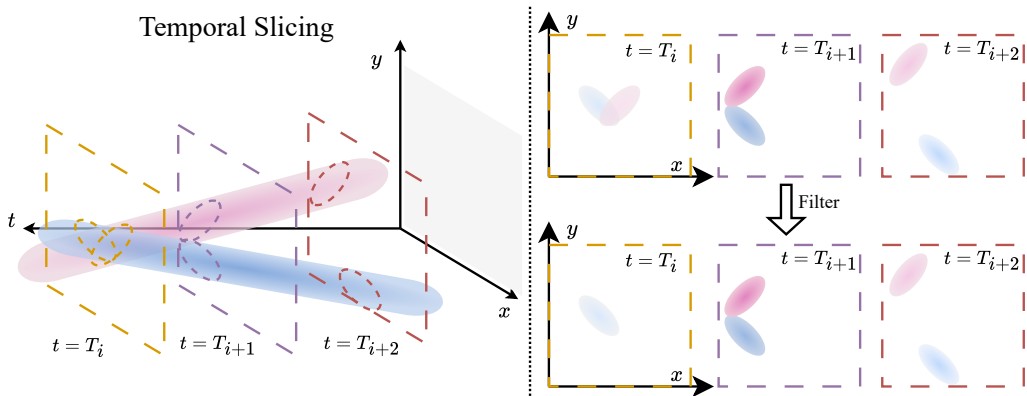

Figure 2: Illustration of temporal slicing in 4DGS, with the z-axis omitted for simplicity. A 4D Gaussian can be conceptualized as a hyper-cylinder in 4D space. Given the specific time query, a corresponding 3D Gaussian ellipsoid is extracted from this hyper-cylinder. The depth of color in the 3D Gaussian ellipsoid represents its temporal opacity. Those 3D Gaussian ellipsoids with temporal opacity below a predefined threshold are excluded from the scene rendering.

3D content production. The emergence of neural radiance field (NeRF) (Mildenhall et al., 2021) enables high-quality novel view synthesis from multi-view image inputs. It has been further extended to represent dynamic scenes by modeling a direct mapping from spatio-temporal coordinates to color and density (Pumarola et al., 2021; Li et al., 2022b; Cao & Johnson, 2023). Despite the impressive visual quality of NeRF-based methods, they require dense sampling along rays, leading to slow rendering speeds that hinder practical applications.

The recent introduction of 3D Gaussian Splatting (3DGS) (Kerbl et al., 2023) marks a significant shift in the field of novel view synthesis. This approach incorporates the explicit 3D Gaussian representation and differentiable tile-based rasterization to enable real-time rendering speeds that significantly outperform NeRF-based methods. Built on this framework, subsequent studies have developed 4D Gaussian Splatting (4DGS) (Yang et al., 2024a; Duan et al., 2024), which conceptualizes scene variations across different timestamps as 4D spatio-temporal Gaussian ellipsoids. As shown in Fig. 2, when depicting a 3D scene at a given timestamp, these 4D Gaussians will first be sliced into 3D Gaussians with time-varying positions and opacity. Then, the 3D Gaussians with the temporal decay opacity below a specific threshold are filtered out. This filtering operation helps 4DGS to describe the transient content such as emerging or vanishing objects. Finally, following 3DGS, the remaining 3D Gaussians are projected onto 2D screens through fast rasterization. By directly optimizing a collection of 4D Gaussians, 4DGS effectively captures both static and dynamic scene elements, thereby achieving photorealistic visual quality.

However, 4DGS requires millions of Gaussians to adequately represent dynamic scenes with high fidelity. As depicted in Fig. 1 (a), rendering the *Birthday* scene necessitates up to 13 million Gaussian points, leading to a storage overhead of approximately 7.79GB. This substantial storage and transmission challenge can severely limit the practical applications of 4DGS, particularly on resource-constrained devices. For example, the significant memory requirements may make it impractical to store, transmit, and render various scenes on AR/VR headsets. Consequently, it is of critical importance to compress 4D Gaussians to minimize the memory footprint of 4DGS while preserving high-quality scene representation and reconstruction.

To address the significant memory and storage challenges associated with 4DGS, we propose a **M**emory-**E**fficient 4D **Ga**ussian Splatting (MEGA) framework. In the original 4D Gaussian representation, 144 out of the total 161 parameters are 4D spherical harmonics (SH) coefficients, which occupy the majority of the storage space and exhibit considerable redundancy. To develop a memory-efficient 4D Gaussian representation, we draw inspiration from the concepts of Direct Current (DC) and Alternating Current (AC) in electrical engineering, which symbolize the steady and varying components, respectively. Specifically, we decouple the color attribute into a per-Gaussian DC color component and a shared temporal-viewpoint aware AC color predictor. This predictor is capable of accurately estimating the color variations of a Gaussian at given times and viewing angles, thereby effectively preserving visual quality. It is noteworthy that our DC color compo-

nent requires only 3 parameters, while the predictor utilizes a lightweight multi-layer perceptron (MLP) with three linear layers. Consequently, this modification achieves a compression ratio of approximately $8\times$ relative to the original 4D Gaussians with equivalent Gaussian points, substantially reducing the storage demands of the Gaussian representation.

Nevertheless, compacting the properties of the 4D Gaussian alone cannot effectively alleviate the problem of excessive number of Gaussians required. Existing 4DGS baselines (Yang et al., 2024a; Duan et al., 2024) assume that each sliced 4D Gaussian exhibits only linear movement over time while maintaining constant covariance, which means that the complex motion in the scene has to be modeled by a combination of multiple Gaussians. Moreover, as illustrated in Fig. 4 (a), only about 6% of Gaussians actively participate in rendering at any given time, because the temporal decay opacity forces each Gaussian to be visible only near its mean time center and invisible at other times. These inherent properties significantly limit the effective utilization of each Gaussian, thereby increasing the number of Gaussians needed for adequate scene rendering. To overcome this limitation, we introduce an efficient entropy-constrained Gaussian deformation field designed to expand the operational range of 4D Gaussians. This deformation model leverages both temporal and viewpoint information to accurately represent Gaussian motion, shape, and transience changes. Meanwhile, a spatial opacity-based entropy loss is introduced to push the spatial opacity of each Gaussian towards binary states (either one or zero). This adjustment aids in identifying and eliminating non-essential Gaussians that contribute minimally to the overall performance. In this way, our proposed strategy not only effectively reduces the number of Gaussians, but also improves the utilization rate of the Gaussians involved in rendering given the time and viewing angle. Finally, to store the parameters of our streamlined 4DGS, we employ 16-bit floating-point (FP16) precision with zip delta compression algorithm to achieve further reductions in memory footprint. In summary, our main contributions are three-fold:

- To the best of our knowledge, we are among the first to develop a memory-efficient framework for 4D Gaussian Splatting. By decomposing the color attribute into a per-Gaussian DC color component and a lightweight, temporal-viewpoint aware AC color predictor, we successfully eliminate the need for redundant spherical harmonics coefficients.

- We introduce an entropy-constrained Gaussian deformation technique to enhance the potential of each 4D Gaussian for depicting complex scene motion. This approach not only substantially reduces the number of Gaussians but also improves their utilization rate. Moreover, we integrate straightforward post-processing techniques, such as FP16 precision and zip delta compression, to further decrease storage overhead.

- Extensive experimental results demonstrate that our proposed method achieves significant storage reductions—approximately $190\times$ and $125\times$ on the Technicolor and Neural 3D Video datasets, respectively—while maintaining comparable quality of scene representation and rendering speed relative to the original 4DGS.

## 2 RELATED WORKS

**Neural Rendering for Static Scenes.** Recently, the advent of neural rendering has attracted increasing interest in 3D scene representation and reconstruction. NeRF, pioneered by Mildenhall et al. (2021), represents the volume density and view-dependent emitted radiance of a 3D scene as a function of 5D coordinates (3D position and 2D viewing direction) using an MLP. However, the vanilla NeRF relies solely on a large MLP to store scene information, significantly limiting its training and rendering efficiency. Subsequent works have explored explicit grid-based representations (Müller et al., 2022; Fridovich-Keil et al., 2022; Chen et al., 2022; Sun et al., 2022) to enhance training efficiency. Nonetheless, these NeRF-based methods still face challenges of slow rendering due to dense sampling for each ray. In contrast, Kerbl et al. (2023) introduce 3D Gaussian Splatting, a novel explicit representation framework that employs a highly optimized custom CUDA rasterizer to achieve unparalleled rendering speeds with high-fidelity novel view synthesis for complex scenes.

**Neural Rendering for Dynamic Scenes.** Synthesizing new views of dynamic scenes from a series of 2D images captured at different times presents a significant challenge. Recent advancements have extended NeRF to handle monocular or multi-object dynamic scenes by learning a mapping from spatio-temporal coordinates to color and density (Lombardi et al., 2019; Mildenhall et al., 2019; Pumarola et al., 2021; Li et al., 2022b;a; Cao & Johnson, 2023; Song et al., 2023; Attal et al.,

2023; Fridovich-Keil et al., 2023; Wang et al., 2023). Unfortunately, these methods suffer from low rendering efficiency. To address this issue, some recent studies (Wu et al., 2024; Yang et al., 2024b; Das et al., 2024; Bae et al., 2024; Lu et al., 2024; Guo et al., 2024) have developed deformable 3D GS, which decouples dynamic scenes into a static canonical 3DGS and a deformation motion field to account for temporal variations in the 3D Gaussian parameters. Concurrently, a series of recent studies (Yang et al., 2024a; Duan et al., 2024; Li et al., 2024; Katsumata et al., 2024; Kratimenos et al., 2024) directly learn a set of spatio-temporal Gaussians to model static, dynamic, and transient content within a scene. However, these methods require a large number of Gaussians to achieve high-quality scene modeling, which brings expensive storage overhead. To this end, our work focuses on developing effective compression techniques for 4DGS (Yang et al., 2024a).

**3D Gaussian Splatting Compression.** Since optimized scenes in 3DGS typically comprise millions of 3D Gaussians and require up to several gigabytes of storage, various compression strategies have been proposed to reduce the size, including redundant Gaussian pruning (Fan et al., 2024; Lee et al., 2024), spherical harmonics distillation or compactness (Lee et al., 2024; Fan et al., 2024; Niedermayr et al., 2024; Wang et al., 2024), vector quantization (Lee et al., 2024; Fan et al., 2024; Wang et al., 2024; Navaneet et al., 2024), and entropy models (Chen et al., 2024). However, due to the differences between 3DGS for static scene representation and 4DGS for dynamic scene representation, existing methods may be inapplicable to or unsuitable for 4DGS. In this paper, we aim to develop a more compact color representation and reduce the number of 4D Gaussians by considering temporal and viewpoint factors, thereby achieving a more efficient memory footprint. As far as we know, our study is among the first in 4DGS compression.

## 3 METHOD

In Section 3.1, we first review the technique of 4DGS (Yang et al., 2024a), which serves as the foundation of our method. Subsequently, in Section 3.2, we introduce how to develop our memory-efficient 4D Gaussian Splatting for modeling dynamic scenes. Finally, we detail the training process and describe how to store our compact 4DGS in Section 3.3.

### 3.1 PRELIMINARY: 4D GAUSSIAN SPLATTING

4D Gaussian Splatting (Yang et al., 2024a) optimizes a set of anisotropic 4D Gaussians via differentiable rasterization to effectively represent a dynamic scene. With a highly efficient rasterizer, the optimized model facilitates real-time rendering of high-fidelity novel views. Each 4D Gaussian is characterized by the following attributes: (i) 4D center $\boldsymbol{\mu}_{4D} = (\mu_x, \mu_y, \mu_z, \mu_t)^T \in \mathbb{R}^4$; (ii) 4D rotation $\boldsymbol{R}_{4D}$ represented by a pair of left quaternion $\boldsymbol{q}_l \in \mathbb{R}^4$ and right quaternion $\boldsymbol{q}_r \in \mathbb{R}^4$; (iii) 4D scaling factor $\boldsymbol{s}_{4D} = (s_x, s_y, s_z, s_t)^T \in \mathbb{R}^4$; (iv) time- and view-dependent RGB color represented by 4D spherical harmonics coefficients $\boldsymbol{h} \in \mathbb{R}^{3(k_v+1)^2(k_t+1)}$ with the view degrees of freedom $k_v$ and time degress of freedom $k_t$; (v) spatial opacity $o \in [0, 1]$.

Given 4D scaling matrix $S_{4D} = diag(\boldsymbol{s}_{4D})$ and 4D rotation matrix $\boldsymbol{R}_{4D}$, we parameterize 4D Gaussian's covariance matrix as:

$$\boldsymbol{\Sigma}_{4D} = \boldsymbol{R}_{4D}\boldsymbol{S}_{4D}\boldsymbol{S}_{4D}^T\boldsymbol{R}_{4D}^T = \begin{pmatrix} \mathbf{U} & \mathbf{V} \\ \mathbf{V}^T & \mathbf{W} \end{pmatrix}, \mathbf{U} \in \mathbb{R}^{3 \times 3}. \tag{1}$$

When rendering the scene at time $t$, each 4D Gaussian is sliced into 3D space. The density of the sliced 3D Gaussian at the spatial point $\boldsymbol{x}$ is expressed as:

$$G_{3D}(\boldsymbol{x}, t) = \sigma(t)e^{-\frac{1}{2}[\boldsymbol{x}-\boldsymbol{\mu}_{3D}(t)]^T \boldsymbol{\Sigma}_{3D}^{-1}[\boldsymbol{x}-\boldsymbol{\mu}_{3D}(t)]}, \tag{2}$$

where $\boldsymbol{\Sigma}_{3D} = \mathbf{U} - \frac{\mathbf{V}\mathbf{V}^T}{\mathbf{W}}$ represents the time-invariant 3D covariance matrix. The temporal decay opacity, $\sigma(t) = e^{-\frac{(t-\mu_t)^2}{2\mathbf{W}}}$, utilizes a 1D Gaussian function to modulate the contribution of each Gaussian to the $t$-th scene. The time-variant 3D center, $\boldsymbol{\mu}_{3D}(t) = \boldsymbol{\mu}_{3D} + (t - \mu_t)\frac{\mathbf{V}}{\mathbf{W}}$, introduces a linear motion term to the 3D center position $\boldsymbol{\mu}_{3D} = (\mu_x, \mu_y, \mu_z)^T$, assuming that all motions can be approximated as linear motion within a very small time range. After temporal slicing, the following process involves projecting sliced 3D Gaussians onto the 2D image plane based on depth order from specific view direction, and executing the fast differentiable rasterization to render the final image.

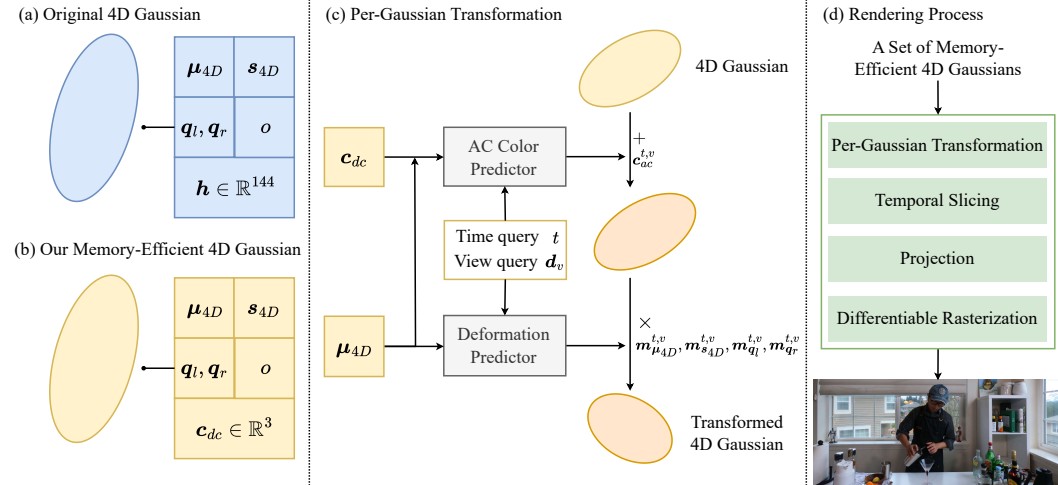

Figure 3: Overview of our proposed memory-efficient Gaussian Splatting framework. (a) The original 4D Gaussian uses 4D spherical harmonics $h$ to represent color, which is highly redundant and consumes substantial memory. (b) Our memory-efficient 4D Gaussian replaces $h$ with a compact, view-independent, and time-independent color component $c_{dc}$, achieving an about $8\times$ reduction in storage overhead. (c) In the per-Gaussian transformation, a lightweight AC color predictor compensates for the absent viewpoint and temporal information in $c_{dc}$, and a deformation predictor expands the action range of each Gaussian. (d) Our rendering process consists of four steps: per-Gaussian transformation, temporal slicing, projection, and differentiable rasterization.

Although this paradigm provides high-quality novel view synthesis, it necessitates large amount of Gaussians to fully reconstruct a dynamic scene, which brings unbearable storage overhead. This challenge drives our memory-efficient 4D Gaussian Splatting design.

## 3.2 MEMORY-EFFICIENT 4D GAUSSIAN SPLATTING FOR DYNAMIC SCENES

**Overview.** As illustrated in Fig. 3, we develop our memory-efficient 4D Gaussian framework to significantly reduce the number of per-Gaussian stored parameters and drive the model to reconstruct dynamic scene with fewer 4D Gaussians. During the rendering process, we utilize a set of optimized 4D Gaussians and initially transform each Gaussian based on specific time and view direction. This transformation procedure involves Gaussian color prediction and geometry deformation. By modifying the geometric structure of each Gaussian, we effectively broaden its action range. This expansion not only reduces the total number of Gaussians required but also increases the rendering participation rate of each Gaussian. Following the per-Gaussian transformation, we adhere to the established protocols of the original 4DGS (Yang et al., 2024a) to carry out temporal slicing, projection, and differentiable rasterization, all critical for rendering high-quality frames.

**Memory-efficient 4D Gaussian.** 4DGS introduces 4D spherical harmonics $h$ to model the temporal evolution of view-dependent color in dynamic scenes, which typically requires 144 of the total 161 parameters and contributes to the main storage overhead. While Lee et al. (2024) have explored the use of a grid-based neural field to replace SH coefficients $h$, we find that directly applying this method results in severe performance loss compared to 4DGS (see Table 3).

To overcome this issue, we propose a compact DC-AC color (DAC) representation. Specifically, we decouple the color attribute as a per-Gaussian DC color component $c_{dc} \in \mathbb{R}^3$ and a temporal-viewpoint aware AC color predictor $\mathcal{F}_\phi$. To predict the final color $c_{t,v}$ of each Gaussian, we first compute the normalized view direction $d_v = \frac{\boldsymbol{\mu}_{3D} - \boldsymbol{p}_v}{||\boldsymbol{\mu}_{3D} - \boldsymbol{p}_v||_2}$ for each Gaussian according to the camera center point $\boldsymbol{p}_v \in \mathbb{R}^3$ at the viewpoint $v$. Then, we concatenate the 3D position $\boldsymbol{\mu}_{3D}$, view direction $d_i$, time $t$, and DC color $c_{dc}$ and input them to a lightweight MLP network $\mathcal{F}_\phi$:

$$c_{t,v} = \text{sigmoid}(c_{dc} + \mathcal{F}_\phi(\text{sg}(\boldsymbol{\mu}_{3D}), \text{sg}(d_v), t, c_{dc})), \qquad (3)$$

where $\text{sg}(\cdot)$ indicates a stop-gradient operation. This hybrid color composition method not only effectively preserves the individual information using DC component and supplements the missing

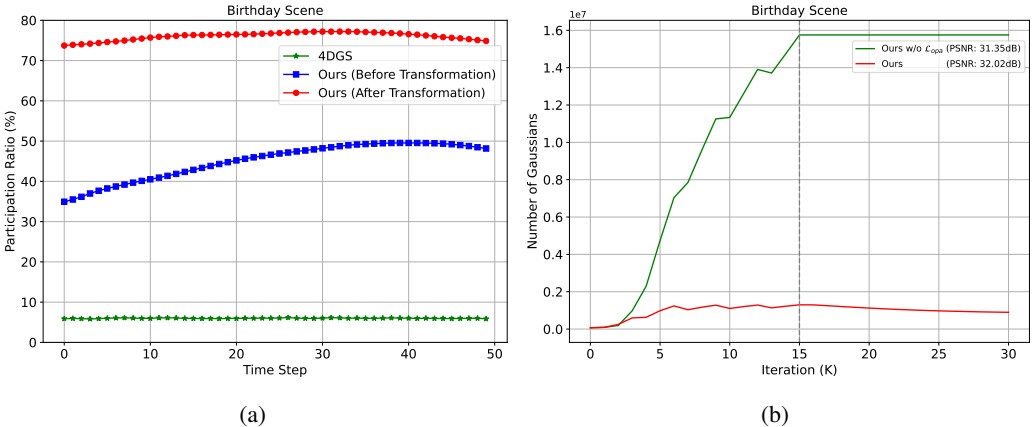

(a)  (b)

Figure 4: (a) The ratio of Gaussians involved in rendering the *Birthday* scene at different time steps. The blue line shows how many Gaussians are involved in rendering in our MEGA model if we do not use per-Gaussian transformation. (b) Visualization of the varying number of Gaussians on the *Birthday* scene during training.

viewpoint and time information using the AC predictor to maintain high rendering quality, but also reduces the storage overhead by up to $8\times$ compared to the original 4DGS (Yang et al., 2024a).

**Entropy-constrained Gaussian Deformation.** For a specific time $t$, 4DGS (Yang et al., 2024a) presupposes that the sliced 4D Gaussians exhibit linear movement while their rotation and scale remain constant. This strict assumption simplifies the movement representation and forces the model to combine multiple extra Gaussians to present any complex non-linear motions. Moreover, the sliced 4D Gaussian introduces the temporal decay opacity $\sigma_t$. From its definition, it is analyzed that a Gaussian gradually appears as time $t$ approaches its temporal position $\mu_t$, peaks in opacity at $t = \mu_t$, and gradually diminishes in density as $t$ moves away from $\mu_t$. As shown in Fig. 4 (a), this limited temporal operation range results in more than 90% of Gaussians being excluded at each time, causing the model to densify a large amount of Gaussians for rendering high-quality scene.

To address these limitations, we advocate for improving flexibility in the motion representation and geometric structure of each 4D Gaussian. Specifically, we introduce a temporal-viewpoint aware deformation predictor to enlarge the action range of Gaussians. The 4D Gaussian center $\boldsymbol{\mu}_{4D}$, view direction $\boldsymbol{d}_i$, and time $t$ are mapped to a high-dimensional space using a regular frequency positional encoding function $\gamma$ (Mildenhall et al., 2021), and then processed through a lightweight MLP network $\mathcal{F}_{\boldsymbol{\theta}}$ to predict the position deformation $\boldsymbol{m}_{\boldsymbol{\mu}_{4D}}^{t,v} \in \mathbb{R}^4$, scale deformation $\boldsymbol{m}_{\boldsymbol{s}_{4D}}^{t,v} \in \mathbb{R}^4$, and rotation deformations $\boldsymbol{m}_{\boldsymbol{q}_l}^{t,v} \in \mathbb{R}^4, \boldsymbol{m}_{\boldsymbol{q}_r}^{t,v} \in \mathbb{R}^4$ as:

$$(\boldsymbol{m}_{\boldsymbol{\mu}_{4D}}^{t,v}, \boldsymbol{m}_{\boldsymbol{s}_{4D}}^{t,v}, \boldsymbol{m}_{\boldsymbol{q}_l}^{t,v}, \boldsymbol{m}_{\boldsymbol{q}_r}^{t,v}) = \mathcal{F}_{\boldsymbol{\theta}}(\gamma(\mathrm{sg}(\boldsymbol{\mu}_{4D})), \gamma(\mathrm{sg}(\boldsymbol{d}_v)), \gamma(t)), \quad (4)$$

where $\gamma$ is defined as $(\sin(2^l\pi p), \cos(2^l\pi p))_{l=0}^{L-1}$. Based on the estimated deformation for time $t$ and viewpoint $v$, we transform the original 4D Gaussian to a temporal-viewpoint aware 4D Gaussian:

$$\boldsymbol{\mu}_{4D}^{t,v} = \boldsymbol{\mu}_{4D} \times \boldsymbol{m}_{\boldsymbol{\mu}_{4D}}^{t,v}, \quad \boldsymbol{s}_{4D}^{t,v} = \boldsymbol{s}_{4D} \times \boldsymbol{m}_{\boldsymbol{s}_{4D}}^{t,v}, \quad \boldsymbol{q}_l^{t,v} = \boldsymbol{q}_l \times \boldsymbol{m}_{\boldsymbol{q}_l}^{t,v}, \quad \boldsymbol{q}_r^{t,v} = \boldsymbol{q}_r \times \boldsymbol{m}_{\boldsymbol{q}_r}^{t,v}. \quad (5)$$

Nonetheless, as depicted in Fig. 4 (b), without constraints on the number of Gaussians, a significant proliferation occurs where Gaussians are continuously split and cloned during the densification process. To force the model to use fewer Gaussians while accurately simulating complex scene changes, we introduce a spatial opacity-based entropy loss $\mathcal{L}_{opa}$ that encourages the spatial opacity $o$ of each Gaussian to approach one or zero:

$$\mathcal{L}_{opa} = \frac{1}{N}\sum_{j=1}^{N}(-o_j \log(o_j)), \quad (6)$$

where $N$ denotes the number of Gaussians. During optimization, we actively prune Gaussians that exhibit near-zero opacity at every $K$ iterations, which ensures efficient computation and maintains a low storage footprint throughout the training phase. Furthermore, as shown in Fig. 4 (a), with the opacity-based entropy loss $\mathcal{L}_{opa}$, our deformation field successfully enlarges the action range of each Gaussian, increasing the Gaussian participation ratio from less than 50% to about 75% under the same Gaussian points.

Table 1: Quantitative comparison with various competitive baselines on the Technicolor Dataset. "Storage" refers to the total model size for 50 frames.

| Method | PSNR↑ | DSSIM$_1$↓ | DSSIM$_2$↓ | LPIPS↓ | FPS↑ | Storage↓ |
|---|---|---|---|---|---|---|
| DyNeRF | 31.80 | - | 0.0210 | 0.1400 | 0.02 | 30.00MB |
| HyperReel | 32.70 | 0.0470 | - | 0.1090 | 4.00 | 60.00MB |
| Deformable 3DGS | 30.95 | 0.0696 | 0.0353 | 0.1553 | 76.09 | 61.36MB |
| STG | 33.35 | 0.0404 | 0.0187 | 0.0846 | 141.73 | 51.35MB |
| E-D3DGS | 32.89 | 0.0494 | 0.0231 | 0.1114 | 79.14 | 56.07MB |
| 4DGS | 32.07 | 0.0535 | 0.0263 | 0.1189 | 55.26 | 6107.07MB |
| Ours | 33.57 | 0.0442 | 0.0204 | 0.1014 | 83.14 | 32.45MB |

## 3.3 Training and Compression Pipeline

**Loss Function.** Following the original 4DGS (Yang et al., 2024a), we adopt the photometric loss $\mathcal{L}_{photo}$, consisting of $\mathcal{L}_1$ loss and structural similarity loss $\mathcal{L}_{ssim}$, to measure the distortion between the rendered image and ground truth image. By adding the loss for opacity regularization $\mathcal{L}_{opa}$, the overall loss $\mathcal{L}$ is defined as:

$$\mathcal{L} = \mathcal{L}_{photo} + \kappa\mathcal{L}_{opa} = (1 - \lambda)\mathcal{L}_1 + \lambda\mathcal{L}_{ssim} + \kappa\mathcal{L}_{opa}, \tag{7}$$

where both $\lambda$ and $\kappa$ are trade-off parameters to balance the components.

**Compression Pipeline.** During the optimization phase, we adopt half-precision training. After obtaining the optimized MEGA representation, we store these learnable parameters in the FP16 format, then apply the zip delta compression algorithm. This lossless compression technique typically reduces storage overhead by approximately 10%.

## 4 Experiments

### 4.1 Experimental Setup

**Datasets.** We evaluate the effectiveness of our method using two real-world benchmarks that are representative of various challenges in dynamic scene rendering: (1) **Technicolor Light Field Dataset** (Sabater et al., 2017): This dataset consists of multi-view video data captured by a time-synchronized 4×4 camera rig. Following HyperReel (Attal et al., 2023), we exclude the camera at the second row, second column and evaluate on five scenes (*Birthday*, *Fabien*, *Painter*, *Theater*, and *Trains*) at 2048×1088 full resolution. (2) **Neural 3D Video Dataset** (Neu3DV) (Li et al., 2022b): This dataset includes six indoor multi-view video scenes captured by 18 to 21 cameras, each at a resolution of 2704×2028 pixels. The scenes (*Coffee Martini*, *Cook Spinach*, *Cut Roasted Beef*, *Flame Salmon*, *Flame Steak*, *Sear Steak*) vary in duration and feature dynamic movements, some with multiple objects in motion. Consistent with existing practices (Li et al., 2022b; Yang et al., 2024a), evaluations are conducted at half resolution of 300-frame scenes.

**Evaluation Metrics.** To assess the quality of rendered videos, we utilize three popular image quality assessment metrics: Peak Signal-to-Noise Ratio (PSNR), Dissimilarity Structural Similarity Index Measure (DSSIM), and Learned Perceptual Image Patch Similarity (LPIPS) (Zhang et al., 2018). PSNR quantifies the pixel color error between the rendered and original frames. DSSIM evaluates the perceived dissimilarity of the rendered image, while LPIPS measures the higher-level perceptual similarity using an AlexNet backbone (Krizhevsky et al., 2012). Given the inconsistency in DSSIM implementation noted across different methods (Fridovich-Keil et al., 2023; Attal et al., 2023), we follow Li et al. (2024) to distinguish DSSIM results into two categories: DSSIM$_1$ and DSSIM$_2$. DSSIM$_1$ is calculated with a data range set to 1.0, based on the structural similarity function from the *scikit-image* library, whereas DSSIM uses a data range of 2.0. For rendering speed, we measure the performance in frames per second (FPS).

**Baselines.** As we introduce MEGA, a novel method for compressing 4DGS (Yang et al., 2024a), our primary comparison focuses on the baseline 4DGS method. Additionally, we benchmark MEGA against a range of NeRF-based baselines, including DyNeRF (Li et al., 2022b), HyperReel (Attal et al., 2023), Neural Volume (Lombardi et al., 2019), LLFF (Mildenhall et al., 2019), HexPlane (Cao

Table 2: Quantitative comparisons with various competitive baselines on the Neural 3D Video Dataset. "Storage" refers to the total model size for 300 frames. [1]: Only report the result on the *Flame Salmon* scene. [2]: Exclude the *Coffee Martini* scene. [3]: These methods train each model with a 50-frame video sequence to prevent memory overflow, requiring six models to complete the overall evaluation.

| Method | PSNR↑ | DSSIM$_1$↓ | DSSIM$_2$↓ | LPIPS↓ | FPS↑ | Storage↓ |
|---|---|---|---|---|---|---|
| Neural Volume[1] | 22.80 | - | 0.0620 | 0.2950 | - | - |
| LLFF[1] | 23.24 | - | 0.0200 | 0.2350 | - | - |
| DyNeRF[1] | 29.58 | - | 0.0200 | 0.0830 | 0.015 | 28.00MB |
| HexPlane[2,3] | 31.71 | - | 0.0140 | 0.0750 | 0.56 | 200.00MB |
| StreamRF | 28.26 | - | - | - | 10.90 | 5310.00MB |
| NeRFPlayer[3] | 30.69 | 0.0340 | - | 0.1110 | 0.05 | 5130.00MB |
| HyperReel | 31.10 | 0.0360 | - | 0.0960 | 2.00 | 360.00MB |
| K-Planes | 31.63 | - | 0.0180 | - | 0.30 | 311.00MB |
| MixVoxels-L | 31.34 | - | 0.0170 | 0.0960 | 37.70 | 500.00MB |
| MixVoxels-X | 31.73 | - | 0.0150 | 0.0640 | 4.60 | 500.00MB |
| Dynamic 3DGS | 30.46 | 0.0350 | 0.0190 | 0.0990 | 460.00 | 2772.00MB |
| C-D3DGS | 30.46 | - | - | 0.1500 | 118.00 | 338.00MB |
| Deformable 3DGS | 30.98 | 0.0331 | 0.0191 | 0.0594 | 29.62 | 32.64MB |
| E-D3DGS | 31.20 | 0.0259 | 0.0151 | 0.0304 | 69.70 | 40.20MB |
| STG[3] | 32.04 | 0.0261 | 0.0145 | 0.0440 | 273.47 | 175.35MB |
| 4DGS | 31.57 | 0.0290 | 0.0164 | 0.0573 | 96.69 | 3128.00MB |
| Ours | 31.49 | 0.0290 | 0.0165 | 0.0568 | 77.42 | 25.05MB |

& Johnson, 2023), NeRFPlayer (Song et al., 2023), MixVoxels (Wang et al., 2023), and K-Planes (Fridovich-Keil et al., 2023). Other recent competitive Gaussian-based methods are also considered in our comparisons, including Dynamic 3DGS (Luiten et al., 2024), C-D3DGS (Katsumata et al., 2024), Deformable 3DGS (Wu et al., 2024), E-D3DGS (Bae et al., 2024), and STG (Li et al., 2024). The numerical results of Deformable 3DGS, E-D3DGS, STG, and 4DGS are produced by running their released codes on a single NVIDIA A800 GPU, while results for other baselines are from their original papers.

**Implementation Details.** We train our MEGA model over 30k iterations and stop densification at the midpoint. We use the Adam optimizer with a batch size of one, adopting the hyperparameter settings from the original 4DGS (Yang et al., 2024a) framework, including loss weight, learning rate, and threshold parameters. When rendering the view at time $t$, we filter out those Gaussians with $\sigma(t) \leq 0.05$. To ensure stable training of our deformation predictor, we introduce weight regularization and set it at $1e^{-6}$. The learning rate of the deformation predictor undergoes exponential decay, starting from $8e^{-4}$ and reducing to $1.6e^{-6}$. For the AC color predictor, we start with an initial learning rate of 0.01, incorporating a 100-step warm-up phase. Subsequently, its learning rate is decreased by a factor of three at the 5k, 15k, and 25k steps. Regarding the hyper-parameters in the loss function, we set $\lambda$ and $\kappa$ as 0.2 and 0.0005, respectively, to balance the contributions of different components.

## 4.2 EXPERIMENTAL RESULTS

Table 1 details a quantitative evaluation of our MEGA method on the Technicolor dataset. Notably, our method surpasses the main baseline 4DGS (Yang et al., 2024a), with PSNR, DSSIM$_1$, DSSIM$_2$, and LPIPS improvements by 1.2dB, 0.01, 0.006, and 0.018, respectively. Meanwhile, it significantly reduces storage requirements, achieving a 190× compactness and improving rendering speed by 50%. When compared with the NeRF-based method HyperReel (Attal et al., 2023), MEGA achieves a substantial improvement in representation, with an increase of about 0.87dB in PSNR and a 20× faster rendering speed, while halving the storage overhead. Moreover, our MEGA records a 0.22dB gain in visual fidelity over the state-of-the-art (SOTA) Gaussian-based method STG (Li et al., 2024), and reduces storage overhead by 40%. Fig. 5 offers qualitative comparisons for the *Theater* and *Painter* scenes, demonstrating that our results contain more vivid details and provide artifact-less rendering. More visual comparisons are available in Appendix A.

Table 3: Ablation study of the proposed components. $N$ denotes the number of Gaussians. The last row represents our final solution.

(a) Technicolor Dataset

| Variants | Birthday | | | | Fabien | | | |
|---|---|---|---|---|---|---|---|---|
| | PSNR↑ | DSSIM$_1$↓ | $N$ ↓ | Params↓ | PSNR↑ | DSSIM$_1$↓ | $N$ ↓ | Params↓ |
| 4DGS (Yang et al., 2024a) | 31.00 | 0.0383 | 13.00M | 2093.56M | 33.57 | 0.0582 | 5.43M | 874.14M |
| w/ grid (Lee et al., 2024) | 30.49 | 0.0410 | 16.33M | 293.07M | 32.99 | 0.0620 | 4.61M | 93.77M |
| w/ DAC | 31.60 | 0.0355 | 15.43M | 308.65M | 34.21 | 0.0587 | 4.57M | 91.48M |
| w/ DAC+Deformation | 31.35 | 0.0368 | 15.75M | 315.36M | 33.02 | 0.0604 | 11.56M | 231.53M |
| w/ DAC+$\mathcal{L}_{opa}$ | 31.46 | 0.0370 | 9.15M | 183.23M | 33.96 | 0.0603 | 2.32M | 46.40M |
| w/ DAC+Deformation+$\mathcal{L}_{opa}$ | 32.02 | 0.0309 | 0.91M | 18.48M | 34.89 | 0.0597 | 0.31M | 6.43M |

(b) Neural 3D Video Dataset

| Variants | Flame Steak | | | | Sear Steak | | | |
|---|---|---|---|---|---|---|---|---|
| | PSNR↑ | DSSIM$_1$↓ | $N$ ↓ | Params↓ | PSNR↑ | DSSIM$_1$↓ | $N$ ↓ | Params↓ |
| 4DGS (Yang et al., 2024a) | 33.19 | 0.0204 | 5.17M | 831.88M | 33.44 | 0.0204 | 3.52M | 567.30M |
| w/ grid (Lee et al., 2024) | 31.07 | 0.0279 | 4.82M | 97.35M | 31.313 | 0.0281 | 3.25M | 70.76M |
| w/ DAC | 33.34 | 0.0210 | 5.31M | 106.33M | 33.67 | 0.0206 | 3.61M | 72.18M |
| w/ DAC+Deformation | 33.47 | 0.0209 | 6.34M | 127.16M | 33.46 | 0.0208 | 4.17M | 83.78M |
| w/ DAC+$\mathcal{L}_{opa}$ | 33.45 | 0.0208 | 2.76M | 55.22M | 33.58 | 0.0215 | 1.99M | 39.74M |
| w/ DAC+Deformation+$\mathcal{L}_{opa}$ | 32.27 | 0.0242 | 0.87M | 17.79M | 33.67 | 0.0200 | 0.56M | 11.50M |

Besides, we report the quantitative comparisons on the Neu3DV dataset in Table 2. Relative to 4DGS, our method achieves up to a $125\times$ compression ratio while preserving similar visual quality and rendering speed. It is observed that compared to the SOTA NeRF-based baseline MixVoxels (Wang et al., 2023), our method achieves a $20\times$ storage reduction and a $16\times$ inference speed improvement, maintaining comparable rendering quality. Furthermore, our approach exhibits higher rendering quality and smaller storage overhead compared to most Gaussian-based methods.

### 4.3 ABLATION STUDY

To validate the effectiveness of various components within our proposed method, we conduct ablation experiments on selected scenes from two datasets. We analyze the impact of these components on scenes from the Technicolor dataset (*Birthday*, *Fabien*) and the Neu3DV dataset (*Flame Steak*, *Sear Steak*). Detailed results are presented in Table 3.

**Compact DC-AC Color Representation.** Building on the original 4DGS, we substitute the 4D SH coefficients with a grid-based neural field representation (Lee et al., 2024), and our proposed DAC representation, respectively. While the grid-based approach, referred to as "w/ grid," achieves a reduction of approximately $10\times$ in parameters, it leads to a significant performance degradation compared to 4DGS. This performance loss may be attributed to the grid's inability to retain sufficient detail, thereby discarding critical information. To address this issue, we use a DC component to preserve essential color information inherently present in the scene, and an AC predictor to encode the temporal-viewpoint variations in color. This method allows us to achieve a comparable reduction in storage as the grid-based approach while maintaining high-quality rendering consistent with 4DGS.

**Entropy-constrained Gaussian Deformation.** This part of our ablation study evaluates the impact of Gaussian deformation and opacity-based entropy loss $\mathcal{L}_{opa}$. Starting from the configuration "w/ DAC", we observe that implementing a deformation predictor alone (referred to as "w/ DAC+Deformation") leads to an increased number of Gaussians. Conversely, employing $\mathcal{L}_{opa}$ without the deformation predictor (referred to as "w/ DAC+$\mathcal{L}_{opa}$") limits the action range of each Gaussian, inhibiting their efficacy. However, when combining our deformation predictor with $\mathcal{L}_{opa}$, this strategy significantly reduces the number of Gaussians needed while maintaining rendering quality comparable to that of 4DGS.

## 5 CONCLUSION

In this paper, we develop a novel, memory-efficient framework tailored for 4D Gaussian Splatting. By decomposing the color attribute into a per-Gaussian direct current component and a shared, lightweight alternating current color predictor, our approach significantly reduces the per-Gaussian parameters without compromising performance. Furthermore, to reduce redundancy among the 4D

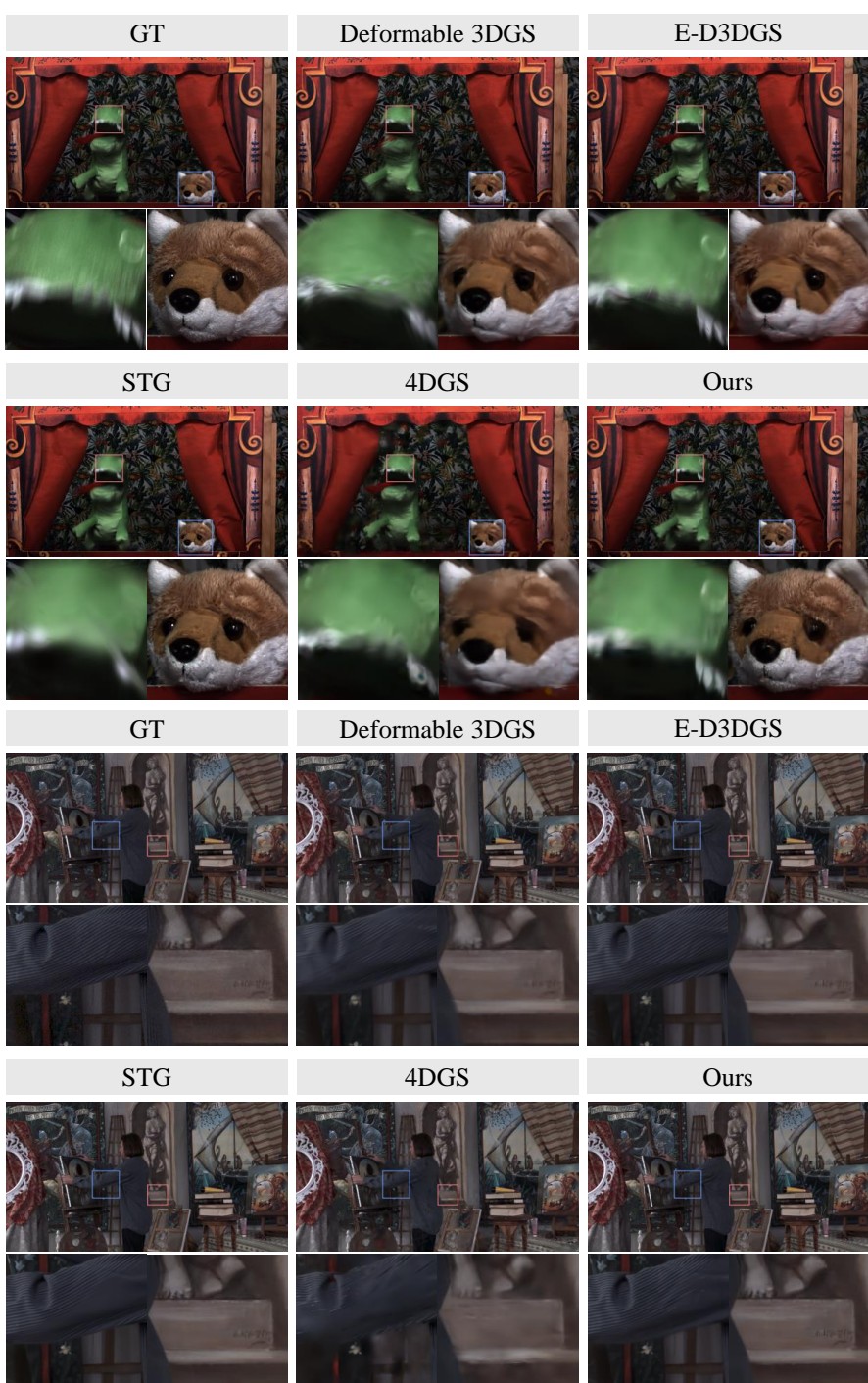

Figure 5: Subjective comparison of various methods on *Theater* scene (Top) and *Painter* scene (Bottom) from the Technicolor Dataset.

Gaussians, we introduce entropy-constrained Gaussian deformation. This technique expands the action range of each Gaussian to enhance the effective utilization rate, thereby enabling the model to render high-quality scenes with as few Gaussians as possible. Extensive experimental results underscore the efficacy of our approach, demonstrating more than a hundredfold reduction in storage requirements while maintaining high-quality reconstruction and real-time rendering speeds in comparison to the original 4D Gaussian Splatting. These advancements establish a new benchmark in the field, combining high performance, compactness, and real-time rendering capabilities.

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

Table 4: Quantitative comparisons with various competitive baselines on the Technicolor Dataset.

| | Birthday | | | | | | Fabien | | | | | |
|---|---|---|---|---|---|---|---|---|---|---|---|---|
| Method | PSNR↑ | DSSIM$_1$↓ | DSSIM$_2$↓ | LPIPS↓ | FPS↑ | Storage↓ | PSNR↑ | DSSIM$_1$↓ | DSSIM$_2$↓ | LPIPS↓ | FPS↑ | Storage↓ |
| DyNeRF | 29.20 | - | 0.0240 | 0.0668 | - | - | 32.76 | - | 0.0175 | 0.2417 | - | - |
| HyperReel | 29.99 | 0.0390 | - | 0.0531 | - | - | 34.70 | 0.0525 | - | 0.1864 | - | - |
| Deformable 3DGS | 30.68 | 0.0440 | 0.0237 | 0.0775 | 52.83 | 90.61MB | 33.33 | 0.0673 | 0.0273 | 0.1851 | 95.52 | 42.81MB |
| E-D3DGS | 31.88 | 0.0328 | 0.0172 | 0.0506 | 62.41 | 66.50MB | 34.69 | 0.0612 | 0.0236 | 0.1689 | 124.71 | 20.02MB |
| STG | 31.65 | 0.0293 | 0.0156 | 0.0413 | 128.43 | 51.81MB | 35.61 | 0.0468 | 0.0177 | 0.1140 | 138.03 | 40.23MB |
| 4DGS | 31.00 | 0.0383 | 0.0211 | 0.0629 | 39.61 | 7986.31MB | 33.57 | 0.0582 | 0.0226 | 0.1555 | 87.54 | 3334.57MB |
| Ours | 32.02 | 0.0309 | 0.0163 | 0.0460 | 61.26 | 31.43MB | 34.89 | 0.0597 | 0.0233 | 0.1760 | 147.58 | 10.26MB |

| | Painter | | | | | | Theater | | | | | |
|---|---|---|---|---|---|---|---|---|---|---|---|---|
| Method | PSNR↑ | DSSIM$_1$↓ | DSSIM$_2$↓ | LPIPS↓ | FPS↑ | Storage↓ | PSNR↑ | DSSIM$_1$↓ | DSSIM$_2$↓ | LPIPS↓ | FPS↑ | Storage↓ |
| DyNeRF | 35.95 | - | 0.0140 | 0.1464 | - | - | 29.53 | - | 0.0305 | 0.1881 | - | - |
| HyperReel | 35.91 | 0.0385 | - | 0.1173 | - | - | 33.32 | 0.0525 | - | 0.1154 | - | - |
| Deformable 3DGS | 34.71 | 0.0497 | 0.0211 | 0.1302 | 84.37 | 51.56MB | 29.65 | 0.0768 | 0.0382 | 0.1795 | 80.40 | 54.75MB |
| E-D3DGS | 35.97 | 0.0360 | 0.0149 | 0.0903 | 94.91 | 38.00MB | 31.04 | 0.0643 | 0.0307 | 0.1493 | 56.88 | 77.61MB |
| STG | 35.73 | 0.0369 | 0.0148 | 0.0963 | 157.01 | 54.84MB | 31.16 | 0.0595 | 0.0286 | 0.1332 | 137.48 | 48.52MB |
| 4DGS | 35.73 | 0.0423 | 0.0176 | 0.1125 | 54.73 | 5667.79MB | 31.29 | 0.0696 | 0.0341 | 0.1653 | 54.05 | 5770.69MB |
| Ours | 36.73 | 0.0380 | 0.0154 | 0.1014 | 121.72 | 14.03MB | 31.54 | 0.0622 | 0.0297 | 0.1475 | 56.91 | 34.31MB |

| | Trains | | | | | | Average | | | | | |
|---|---|---|---|---|---|---|---|---|---|---|---|---|
| Method | PSNR↑ | DSSIM$_1$↓ | DSSIM$_2$↓ | LPIPS↓ | FPS↑ | Storage↓ | PSNR↑ | DSSIM$_1$↓ | DSSIM$_2$↓ | LPIPS↓ | FPS↑ | Storage↓ |
| DyNeRF | 31.58 | - | 0.0190 | 0.0670 | - | - | 31.80 | - | 0.0210 | 0.1400 | 0.02 | 30.00MB |
| HyperReel | 29.74 | 0.0525 | - | 0.0723 | - | - | 32.70 | 0.0470 | - | 0.1090 | 4.00 | 60.00MB |
| Deformable 3DGS | 26.39 | 0.1104 | 0.0663 | 0.2040 | 67.32 | 67.08MB | 30.95 | 0.0696 | 0.0353 | 0.1553 | 76.09 | 61.36MB |
| E-D3DGS | 30.87 | 0.0525 | 0.0289 | 0.0976 | 56.81 | 78.23MB | 32.89 | 0.0494 | 0.0231 | 0.1114 | 79.14 | 56.07MB |
| STG | 32.61 | 0.0296 | 0.0169 | 0.0380 | 147.70 | 61.34MB | 33.35 | 0.0404 | 0.0187 | 0.0846 | 141.73 | 51.35MB |
| 4DGS | 28.79 | 0.0590 | 0.0362 | 0.0985 | 40.36 | 7775.97MB | 32.07 | 0.0535 | 0.0263 | 0.1189 | 55.26 | 6107.07MB |
| Ours | 32.69 | 0.0301 | 0.0172 | 0.0362 | 28.25 | 72.21MB | 33.57 | 0.0442 | 0.0204 | 0.1014 | 83.14 | 32.45MB |

# A  EXPERIMENTAL RESULTS

We provide the complete results on the Technicolor and Neural 3D Video datasets in Table 4 and Table 5. More visualizations are available in Fig. 6 and Fig. 7.

# B  NETWORK STRUCTURE

**AC Color Predictor.** Fig. 8 (a) shows the details of the AC color predictor. After generating the AC color component $c_{ac}^{t,v}$, we combine the DC component $c_{dc}$ to produce the final color $c_{t,v}$.

**Deformation Predictor.** Fig. 8 (b) provides the details of the deformation predictor. For the feature fusion module, we apply two linear layers with ReLU activation function.

Table 5: Quantitative comparisons with various competitive baselines on the Neural 3D Video Dataset. [1]: Only report the result on the *Flame Salmon* scene. [2]: Exclude the *Coffee Martini* scene. [3]: These methods train each model with a 50-frame video sequence to prevent memory overflow, requiring six models to complete the overall evaluation. [4]: Only report the overall results.

### Coffee Martini

| Method | PSNR↑ | DSSIM$_1$↓ | DSSIM$_2$↓ | LPIPS↓ | FPS↑ | Storage↓ |
|---|---|---|---|---|---|---|
| HexPlane[2,3] | - | - | - | - | - | - |
| NeRFPlayer[3] | 31.53 | 0.0245 | - | 0.085 | - | - |
| HyperReel | 28.37 | 0.0540 | - | 0.1270 | - | - |
| K-Planes | 29.99 | - | 0.0170 | - | - | - |
| MixVoxels-L | 29.63 | - | 0.0162 | 0.099 | - | - |
| MixVoxels-X | 30.39 | - | 0.0160 | 0.062 | - | - |
| Dynamic 3DGS | 26.49 | 0.0263 | 0.0129 | 0.087 | - | - |
| Deformable 3DGS | 27.88 | 0.0470 | 0.0284 | 0.0855 | 26.89 | 33.84MB |
| E-D3DGS | 29.56 | 0.0319 | 0.0193 | 0.0300 | 51.94 | 57.97MB |
| STG[3] | 28.55 | 0.0418 | 0.0253 | 0.0692 | 221.76 | 214.52MB |
| 4DGS | 27.98 | 0.0435 | 0.0265 | 0.0847 | 78.79 | 3704.58MB |
| Ours | 27.84 | 0.0440 | 0.0270 | 0.0770 | 75.66 | 24.90MB |

### Cook Spinach

| Method | PSNR↑ | DSSIM$_1$↓ | DSSIM$_2$↓ | LPIPS↓ | FPS↑ | Storage↓ |
|---|---|---|---|---|---|---|
| HexPlane[2,3] | 32.04 | - | 0.0150 | 0.0820 | - | - |
| NeRFPlayer[3] | 30.56 | 0.0355 | - | 0.1130 | - | - |
| HyperReel | 32.30 | 0.0295 | - | 0.0890 | - | - |
| K-Planes | 31.82 | - | 0.0170 | - | - | - |
| MixVoxels-L | 32.40 | - | 0.0157 | 0.088 | - | - |
| MixVoxels-X | 32.63 | - | 0.0146 | 0.057 | - | - |
| Dynamic 3DGS | 30.72 | 0.0295 | 0.0161 | 0.090 | - | - |
| Deformable 3DGS | 33.06 | 0.0267 | 0.0142 | 0.0519 | 31.06 | 33.21MB |
| E-D3DGS | 32.71 | 0.0219 | 0.0123 | 0.0255 | 74.11 | 36.82MB |
| STG[3] | 33.18 | 0.0215 | 0.0113 | 0.0367 | 290.03 | 151.52MB |
| 4DGS | 32.73 | 0.0245 | 0.0133 | 0.0489 | 111.77 | 2474.94MB |
| Ours | 33.08 | 0.0230 | 0.0125 | 0.0471 | 92.51 | 19.83MB |

### Cut Roasted Beef

| Method | PSNR↑ | DSSIM$_1$↓ | DSSIM$_2$↓ | LPIPS↓ | FPS↑ | Storage↓ |
|---|---|---|---|---|---|---|
| Neural Volume[1] | - | - | - | - | - | - |
| LLFF[1] | - | - | - | - | - | - |
| DyNeRF[1] | - | - | - | - | - | - |
| HexPlane[2,3] | 32.55 | - | 0.0130 | 0.0800 | - | - |
| NeRFPlayer[3] | 29.35 | 0.0460 | - | 0.1440 | - | - |
| HyperReel | 32.92 | 0.0275 | - | 0.084 | - | - |
| K-Planes | 31.82 | - | 0.0170 | - | - | - |
| MixVoxels-L | 32.40 | - | 0.0157 | 0.088 | - | - |
| MixVoxels-X | 32.63 | - | 0.0146 | 0.057 | - | - |
| Dynamic 3DGS | 30.72 | 0.0295 | 0.0161 | 0.0900 | - | - |
| Deformable 3DGS | 31.43 | 0.0333 | 0.0204 | 0.0551 | 28.43 | 33.14MB |
| E-D3DGS | 33.02 | 0.0213 | 0.0116 | 0.0258 | 74.33 | 36.63MB |
| STG[3] | 33.55 | 0.0207 | 0.0106 | 0.0367 | 299.98 | 135.28MB |
| 4DGS | 33.23 | 0.0226 | 0.0119 | 0.0470 | 109.11 | 2555.56MB |
| Ours | 33.58 | 0.0217 | 0.0113 | 0.0489 | 75.22 | 25.20MB |

### Flame Salmon

| Method | PSNR↑ | DSSIM$_1$↓ | DSSIM$_2$↓ | LPIPS↓ | FPS↑ | Storage↓ |
|---|---|---|---|---|---|---|
| Neural Volume[1] | 22.80 | - | 0.0620 | 0.2950 | - | - |
| LLFF[1] | 23.24 | - | 0.0200 | 0.2350 | - | - |
| DyNeRF[1] | 29.58 | - | 0.0200 | 0.0830 | 0.015 | 28.00MB |
| HexPlane[2,3] | 29.47 | - | 0.0180 | 0.0780 | - | - |
| NeRFPlayer[3] | 31.65 | 0.0300 | - | 0.098 | - | - |
| HyperReel | 28.26 | 0.0590 | - | 0.136 | - | - |
| K-Planes | 30.44 | - | 0.0235 | - | - | - |
| MixVoxels-L | 29.81 | - | 0.0255 | 0.116 | - | - |
| MixVoxels-X | 30.60 | - | 0.0233 | 0.078 | - | - |
| Dynamic 3DGS | 26.92 | 0.0512 | 0.0302 | 0.1220 | - | - |
| Deformable 3DGS | 28.70 | 0.0432 | 0.0255 | 0.0804 | 28.72 | 34.17MB |
| E-D3DGS | 29.79 | 0.0363 | 0.0216 | 0.0535 | 61.03 | 45.08MB |
| STG[3] | 29.48 | 0.0375 | 0.0224 | 0.0630 | 215.69 | 268.39MB |
| 4DGS | 28.86 | 0.0425 | 0.0257 | 0.0832 | 64.31 | 4695.46MB |
| Ours | 28.4802 | 0.0412 | 0.0251 | 0.0251 | 64.07 | 30.26MB |

### Flame Steak

| Method | PSNR↑ | DSSIM$_1$↓ | DSSIM$_2$↓ | LPIPS↓ | FPS↑ | Storage↓ |
|---|---|---|---|---|---|---|
| HexPlane[2,3] | 32.08 | - | 0.0110 | 0.0660 | - | - |
| NeRFPlayer[3] | 31.93 | 0.0250 | - | 0.0880 | - | - |
| HyperReel | 32.20 | 0.0255 | - | 0.078 | - | - |
| K-Planes | 32.38 | - | 0.0150 | - | - | - |
| MixVoxels-L | 31.83 | - | 0.0144 | 0.088 | - | - |
| MixVoxels-X | 32.10 | - | 0.0137 | 0.051 | - | - |
| Dynamic 3DGS | 33.24 | 0.0233 | 0.0113 | 0.0790 | - | - |
| Deformable 3DGS | 31.83 | 0.0248 | 0.0137 | 0.0418 | 30.91 | 30.72MB |
| E-D3DGS | 30.23 | 0.0241 | 0.0149 | 0.0243 | 76.92 | 32.244MB |
| STG[3] | 33.59 | 0.0178 | 0.0088 | 0.0290 | 305.22 | 141.25MB |
| 4DGS | 33.19 | 0.0204 | 0.0106 | 0.0389 | 91.52 | 3173.37MB |
| Ours | 32.27 | 0.0242 | 0.0129 | 0.0538 | 63.84 | 30.48MB |

### Sear Steak

| Method | PSNR↑ | DSSIM$_1$↓ | DSSIM$_2$↓ | LPIPS↓ | FPS↑ | Storage↓ |
|---|---|---|---|---|---|---|
| HexPlane[2,3] | 32.39 | - | 0.0110 | 0.0700 | - | - |
| NeRFPlayer[3] | 29.13 | 0.0460 | - | 0.138 | - | - |
| HyperReel | 32.57 | 0.0240 | - | 0.077 | - | - |
| K-Planes | 32.52 | - | 0.0130 | - | - | - |
| MixVoxels-L | 32.10 | - | 0.0122 | 0.080 | - | - |
| MixVoxels-X | 32.33 | - | 0.0121 | 0.053 | - | - |
| Dynamic 3DGS | 33.68 | 0.0224 | 0.0105 | 0.079 | - | - |
| Deformable 3DGS | 33.01 | 0.0237 | 0.0125 | 0.0416 | 31.73 | 30.74MB |
| E-D3DGS | 31.91 | 0.0200 | 0.0110 | 0.0233 | 79.89 | 32.426MB |
| STG[3] | 33.89 | 0.0174 | 0.0085 | 0.0295 | 308.15 | 141.16MB |
| 4DGS | 33.44 | 0.0204 | 0.0105 | 0.0411 | 124.66 | 2164.07MB |
| Ours | 33.67 | 0.0200 | 0.0103 | 0.0403 | 93.21 | 19.62MB |

### Average

| Method | PSNR↑ | DSSIM$_1$↓ | DSSIM$_2$↓ | LPIPS↓ | FPS↑ | Storage↓ |
|---|---|---|---|---|---|---|
| Neural Volume[1] | 22.80 | - | 0.0620 | 0.2950 | - | - |
| LLFF[1] | 23.24 | - | 0.0200 | 0.2350 | - | - |
| DyNeRF[1] | 29.58 | - | 0.0200 | 0.0830 | 0.015 | 28.00MB |
| HexPlane[2,3] | 31.71 | - | 0.0140 | 0.0750 | 0.56 | 200.00MB |
| StreamRF[4] | 28.26 | - | - | - | 10.90 | 5310.00MB |
| NeRFPlayer[3] | 30.69 | 0.0340 | - | 0.1110 | 0.05 | 5130.00MB |
| HyperReel | 31.10 | 0.0360 | - | 0.0960 | 2.00 | 360.00MB |
| K-Planes | 31.63 | - | 0.0180 | - | 0.30 | 311.00MB |
| MixVoxels-L | 31.34 | - | 0.0170 | 0.0960 | 37.70 | 500.00MB |
| MixVoxels-X | 31.73 | - | 0.0150 | 0.0640 | 4.60 | 500.00MB |
| Dynamic 3DGS | 30.46 | 0.0350 | 0.0190 | 0.0990 | 460.00 | 2772.00MB |
| C-D3DGS[4] | 30.46 | - | - | 0.1500 | 118.00 | 338.00MB |
| Deformable 3DGS | 30.98 | 0.0331 | 0.0191 | 0.0594 | 29.62 | 32.64MB |
| E-D3DGS | 31.20 | 0.0259 | 0.0151 | 0.0304 | 69.70 | 40.20MB |
| STG[3] | 32.04 | 0.0261 | 0.0145 | 0.0440 | 273.47 | 175.35MB |
| 4DGS | 31.57 | 0.0290 | 0.0164 | 0.0573 | 96.69 | 3128.00MB |
| Ours | 31.49 | 0.0290 | 0.0165 | 0.0568 | 77.42 | 25.05MB |

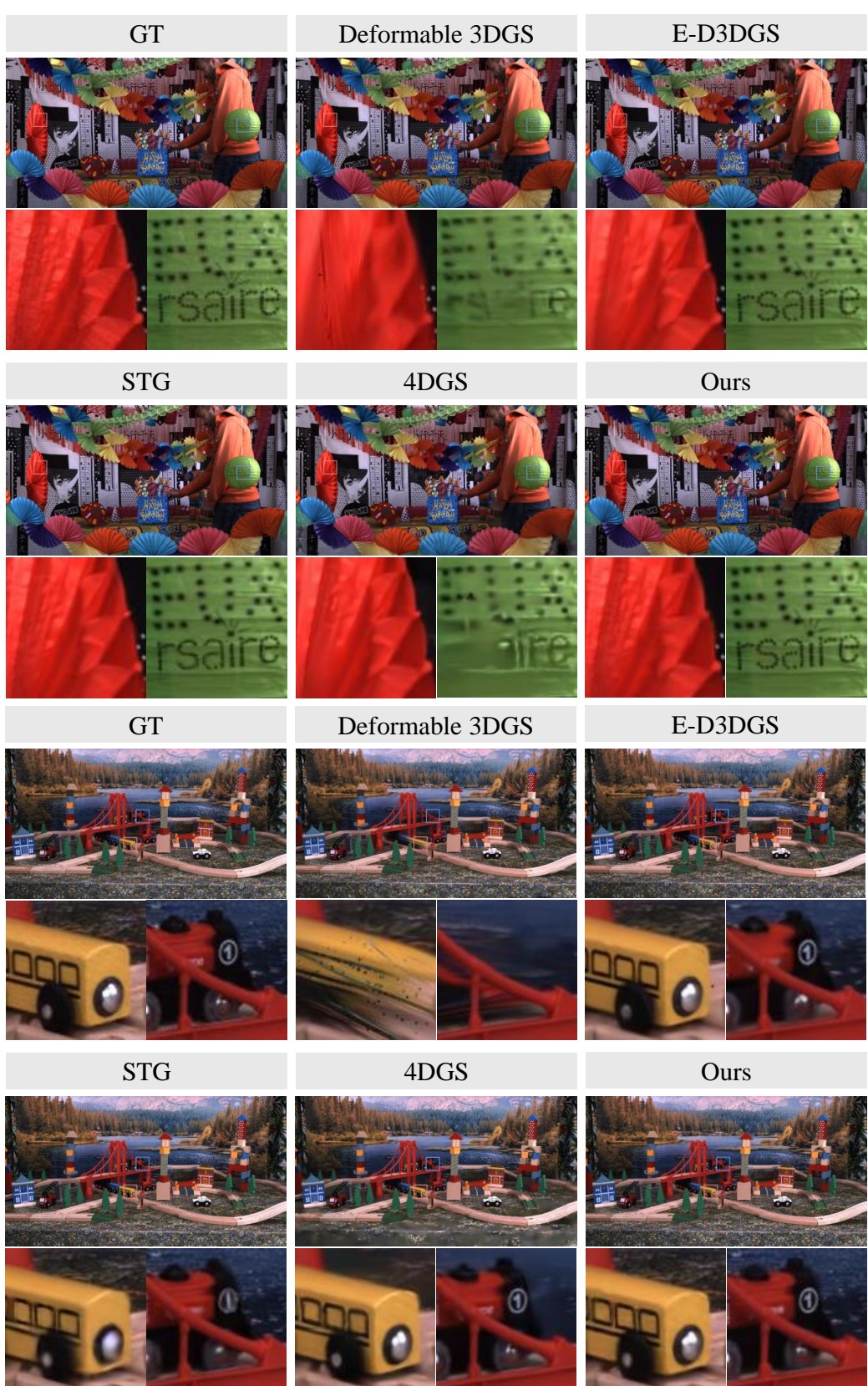

Figure 6: Subjective comparison of various methods on *Birthday* scene (Top) and *Trains* scene (Bottom) from the Technicolor Dataset.

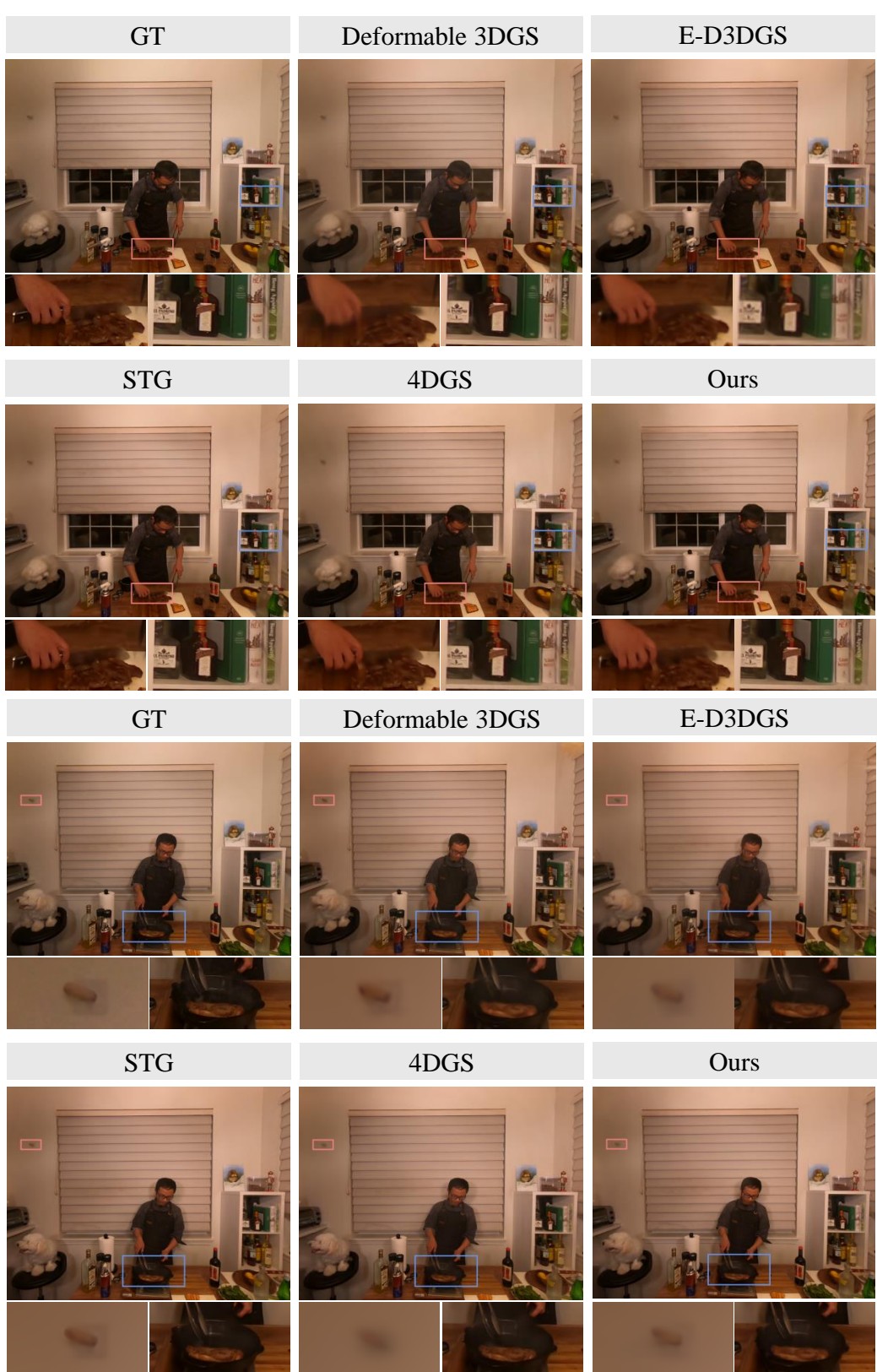

Figure 7: Subjective comparison of various methods on *Cut Roasted Beef* scene (Top) and *Sear Steak* scene (Bottom) from the Neural 3D Video Dataset.

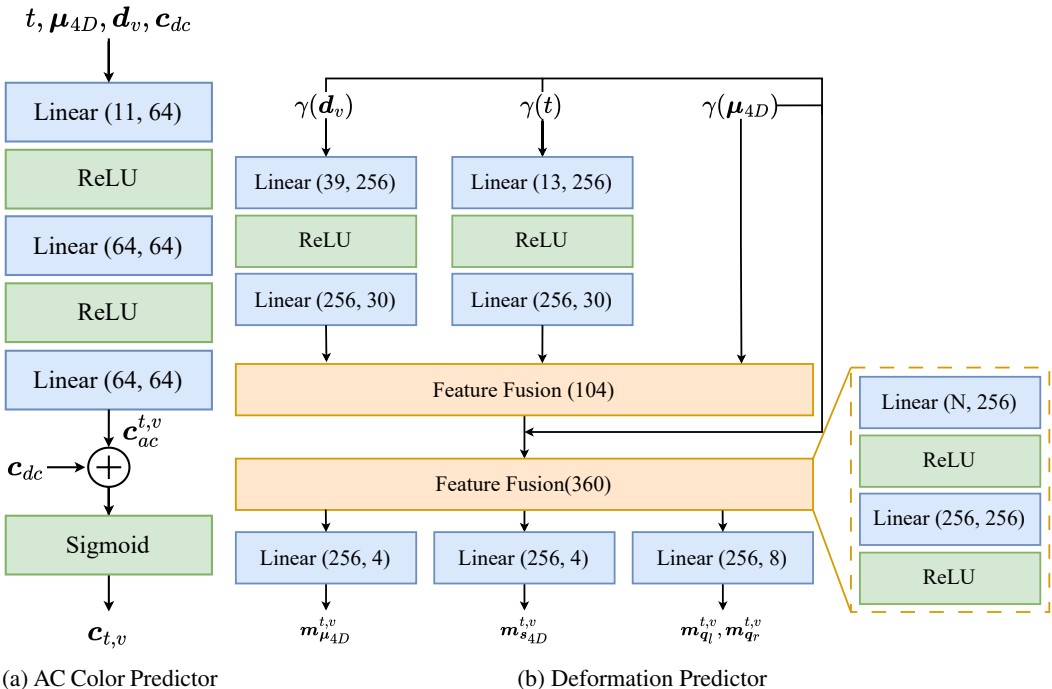

Figure 8: The network structures of (a) AC color predictor, (b) Deformation predictor.

