# OpenReview forum: "MEGA: Memory-Efficient 4D Gaussian Splatting for Dynamic Scenes"
_ICLR.cc/2025/Conference — ICLR 2025 Conference Withdrawn Submission_

### Official Review · Reviewer_RU8T · 2024-10-17

**Soundness:** 2
**Presentation:** 3
**Contribution:** 2
**Rating:** 3
**Confidence:** 5

**Summary:**

The paper introduces a memory-efficient framework for 4D Gaussian Splatting (4DGS), which significantly reduces the memory requirements for dynamic scene reconstruction. It streamlines the color attribute by decomposing it into a direct color component and a lightweight alternating current color predictor, effectively eliminating the need for spherical harmonics coefficients. It also uses an entropy-constrained Gaussian deformation technique, which increases each Gaussian's action range and ensures high rendering quality while using fewer Gaussians.

**Strengths:**

1. The paper's novel approach to reducing memory and computational overhead in dynamic scene rendering is innovative.
2. The paper demonstrates substantial improvements in memory usage and processing speed without compromising the quality of scene representation.
3. Its motivation is well stated and the paper is logical.

**Weaknesses:**

1. The paper could benefit from a more detailed discussion of the potential limitations or scenarios where the approach may not perform optimally.
2. The modifications to the spherical harmonics functions in the model may adversely affect the rendering quality of scenes with metallic surfaces and complex lighting conditions. This could limit the method's effectiveness in accurately representing such materials and lighting scenarios.

**Questions:**

1. Could you provide some visual results in the ablation study? e.g. visual results comparing the full model to versions without the DC-AC color representation and without the entropy-constrained deformation, focusing on areas where these components are expected to have the most impact.
2. How does the model perform in highly dynamic scenarios(qualitatively and quantitatively)? e.g. rapid camera movement, multiple fast-moving objects, or sudden appearance/disappearance of objects.
3. How does the entropy loss component affect the convergence time? e.g. convergence curves with and without the entropy loss. Also, do you have any hyperparameter tuning needed for the entropy loss to balance convergence speed and final performance?
4. Could you provide some comparison results of some metal and lighting scenes?  For example, visual results on scenes with metallic objects or complex lighting conditions, comparing the proposed method to the original 4DGS and other relevant baselines. (qualitatively and quantitatively)

---

### Official Review · Reviewer_BnG4 · 2024-10-26

**Soundness:** 3
**Presentation:** 3
**Contribution:** 2
**Rating:** 5
**Confidence:** 4

**Summary:**

This paper aims to improve the memory efficiency of 4D Gaussian Splatting. To this end, the authors first replace spherical harmonics, which occupy a significant amount of memory, with much smaller DAC. In addition, the authors propose using a deformation field and opacity-based regularizing loss to better utilize Gaussians. With these methods, the paper demonstrates a significant reduction in size compared to 4DGS while maintaining or improving rendering performance.

**Strengths:**

The paper effectively exploits the fact that a large portion of memory is dedicated to color representations and uses MLPs to generate view dependent colors.
In addition, the paper found that many Gaussians are underutilized in the 4D Gaussian Splatting framework and proposed a method to overcome the issue.

**Weaknesses:**

Separating view-dependent and view-independent components has long been a common practice in computer graphics. While the attempt to associate this concept with Direct and Alternating Currents (DC and AC) is interesting, the connection between DC/AC and color separation is not clear in the paper. Providing a clear explanation of why this terminology is used for color separation would highlight the authors' intention.

In line 131, the paper states that this method is the first memory-efficient framework for 4D Gaussian Splatting. It would be helpful to clarify whether “4D Gaussian Splatting” refers to Gaussian-splatting-based dynamic scene representation methods in general or to a specific 4D Gaussian Splatting approach. If it is the former, the performance improvement appears questionable when compared to other Gaussian-splatting-based dynamic scene representation methods, including STG, especially on non-technicolor datasets like the Neural 3D Video Dataset. If it is the latter, providing more motivation for why optimizing a 4D representation is important compared to other variants like STG would strengthen the paper.

Although the analysis that many Gaussians are underutilized is interesting, the ablation study in Tab 3. does not provide supportive results that the deformation field can improve performance while reducing model size exploiting this underutilized phenomenon. In contrast, it seems it often leads to a drop in representation quality while increasing model sizes, which could make the purpose of this component confusing. I would appreciate more detailed explanations of the rule of this component and analysis of the results, which is currently missing in the ablation study (Section 4.3).

**Questions:**

As I mentioned in the weakness section, Tab. 3 raises a question of the role of the deformation field. I would like to know whether the size reduction in the full model (with DAC, the deformation field, and opacity loss) was due to hyperparameter tuning or if it is a pure output combining the two (deformation field and opacity loss).
That is, it would be helpful to know more details for deciding whether the deformation fields are necessary to achieve a high compression ratio.
If it is the latter, delving into how this phenomenon occurs, such as by adjusting the loss weights for the opacity loss and conducting further analysis, would help readers understand the underlying mechanisms.

---

### Official Review · Reviewer_zkv8 · 2024-11-03

**Soundness:** 2
**Presentation:** 2
**Contribution:** 1
**Rating:** 3
**Confidence:** 4

**Summary:**

This paper proposes a compact 4D Gaussian splatting (4DGS) framework that effectively reduces the number of parameters to represent high-quality 4D scenes. To achieve this, the authors introduce a) entropy-constrained Gaussian deformation design and b) DAC color representation, based on the implementation of 4D Gaussian splatting. Consequently, this method demonstrates comparable (or superior) rendering quality while requiring minimal storage size, compared to existing 4DGS approaches.

**Strengths:**

- The view-adaptive Gaussian deformation leads to a significant reduction in the number of Gaussians while preserving the high rendering quality of 4D scenes.
- Also, the authors demonstrate the effectiveness of DAC color representation that encodes the time-aware color using DC components and AC components.
- The opacity regularization further reduces the number of less-contributing Gaussians.

**Weaknesses:**

- The proposed Gaussian deformation would require a large computation for the 4D Gaussian deformation stage, computation with MLPs that encode much information, for each rendering view and each timestamp. Thus, the rendering speed comparison with a similar number of Gaussian settings of the original 4DGS could help understand the additional computation costs for the proposed deformation design.
- In Table 3, the most critical storage reduction comes from the opacity regularization, which has already been proposed in the previous work. This means that the DAC and proposed 4D Gaussian representation are not the main factors for outperforming existing efficient 4DGS approaches.
- It shows limited performance compared to existing methods, STG and E-D3DGS. Firstly, STG demonstrates a rendering speed of 1.7x to 3.5x of MEGA, as shown in Tables 1 and 2. Also, STG and E-D3DGS show better performance in DSSIM and LPIPS, as shown in Table 2. This indicates that simply reducing the number of STG or E-D3DGS can be a more efficient representation that shows high rendering quality with less storage and faster rendering speed. To prove the proposed deformation design is more efficient than existing methods, it is required to compare MEGA with a reduced-Gaussian version of STG and E-D3DGS. It could be done by applying the opacity regularization term of MEGA to those methods, for achieving a similar storage size of MEGA.

**Questions:**

- Please add more comparisons with STG and E-D3DGS adding opacity regularization with several loss weights. It could be helpful to show the rendering quality of STG and E-D3DGS with similar storage sizes to the proposed MEGA.

- Also, there is no comparison with 4DRotorGS [1] although the authors have mentioned it in the manuscript. Please add the comparison with 4DRotorGS.

- Also, the training time can be an important factor for efficient 4DGS representations. It could be helpful to report the training time.

- Furthermore, to compare with Compact3DGS [2] methodology, it is fairer to compare with its extended version [3], which adds dynamic scene results, for the ablation study. Simply increasing the input dimension of grid-based color representation leads to severe degradation as shown in Table 3.  Thus the time-aware static color and time-aware view-dependent color should be considered separately as introduced in the extended version of Compact3DGS [3]. Although it has not been published yet, just an arXiv paper, it also shows superior performance compared to this work. This is not a mandatory suggestion, but rather a recommendation.

- Finally, this paper lacks technical novelty. The view-adaptive encoding scheme has already been introduced in anchor-based 3DGS approaches, such as Scaffold-GS [4] and HAC [5]. What can be carefully considered for extending view-adaptive rendering to 4D scenes? Additionally, the DAC color representation simply replaces the definition of color representation with an MLP-based network, which has been already proposed in STG, even though the rendering quality of MEGA is inferior to that of STG. Please clarify the effectiveness of DAC color representation compared to STG with an additional ablation study with STG for color representation.

---
**Reference**
1. Duan et al., 4D-Rotor Gaussian Splatting: Towards Efficient Novel View Synthesis for Dynamic Scenes, SIGGRAPH 2024
2. Lee et al., Compact 3D Gaussian Representation for Radiance Field, CVPR 2024
3. Lee et al., Compact 3D Gaussian Splatting for Static and Dynamic Radiance Fields, arXiv 2024
4. Lu et al., Scaffold-GS: Structured 3D Gaussians for View-Adaptive Rendering, CVPR 2024
5. Chen et al., HAC: Hash-grid Assisted Context for 3D Gaussian Splatting Compression, ECCV 2024

---

### Note · Authors · 2024-11-13

I have read and agree with the venue's withdrawal policy on behalf of myself and my co-authors.